# Unleashing the Influence of cAMP Receptor Protein: The Master Switch of Bacteriocin Export in *Pectobacterium carotovorum* subsp. *carotovorum*

**DOI:** 10.3390/ijms24119752

**Published:** 2023-06-05

**Authors:** Chung-Pei Chang, Ruchi Briam James Sersenia Lagitnay, Tzu-Rong Li, Wei-Ting Lai, Reymund Calanga Derilo, Duen-Yau Chuang

**Affiliations:** 1Department of Anesthesiology, Show Chwan Memorial Hospital, Changhua 500, Taiwan; fentanylpei@yahoo.com.tw; 2College of Arts and Sciences, Bayombong Campus, Nueva Vizcaya State University, Bayombong 3700, Philippines; rslagitnay@nvsu.edu.ph; 3Department of Chemistry, National Chung Hsing University, Taichung City 400, Taiwan; yamyrose@gmail.com (T.-R.L.); pppolly425@gmail.com (W.-T.L.); rcderilo@nvsu.edu.ph (R.C.D.); 4College of Teacher Education, Bambang Campus, Nueva Vizcaya State University, Bambang 3702, Philippines

**Keywords:** cAMP receptor protein, low-molecular-weight bacteriocin, flagellar-type III secretion, *Pectobacterium carotovorum* subsp. *carotovorum*

## Abstract

*Pectobacterium carotovorum subsp. carotovorum* (*Pcc*) is a Gram-negative phytopathogenic bacterium that produces carocin, a low-molecular-weight bacteriocin that can kill related strains in response to factors in the environment such as UV exposure or nutritional deficiency. The function of the catabolite activator protein (CAP), also known as the cyclic AMP receptor protein (CRP), as a regulator of carocin synthesis was examined. The *crp* gene was knocked out as part of the investigation, and the outcomes were assessed both in vivo and in vitro. Analysis of the DNA sequence upstream of the translation initiation site of carocin S3 revealed two putative binding sites for CRP that were confirmed using a biotinylated probe pull-down experiment. This study revealed that the deletion of *crp* inhibited genes involved in extracellular bacteriocin export via the flagellar type III secretion system and impacted the production of many low-molecular-weight bacteriocins. The biotinylated probe pull-down test demonstrated that when UV induction was missing, CRP preferentially attached to one of the two CAP sites while binding to both when UV induction was present. In conclusion, our research aimed to simulate the signal transduction system that controls the expression of the carocin gene in response to UV induction.

## 1. Introduction

Carocin is a low-molecular-weight bacteriocin produced by *Pectobacterium carotovorum subsp. carotovorum* (*Pcc*) under the influence of environmental stimulation such as ultraviolet irradiation, glucose, or others [1].

Earlier studies have shown that *Pcc* produces low-molecular-weight bacteriocins such as carocin (S1, S2, S3, and D), pectocin (P, M1, and M2), and carotovorocin, a high-molecular-weight, phage-like bacteriocin with unique characteristics [1,2,3,4,5]. These bacteriocins are known as colicin-like bacteriocins (CLB) [6].

These bacteriocin genes are found on the *Pcc* chromosome [2,3,4], and they are activated to make bacteriocin proteins when the bacterial growth environment shifts, suggesting a risk to the bacterium. It is apparent that a pathogen needs metabolic processes to initiate an infection and that these processes adapt to nutritional situations while an infection is active [7]. Many pathogens are stimulated to produce virulence factors by environmental characteristics suggestive of a host environment [8]. Additionally, virulence features are significantly influenced by global regulatory networks, such as the stringent response system, which reacts to a variety of nutritional and metabolic challenges, or the carbon catabolite repression and carbon storage regulator systems, respectively [9,10,11,12,13].

In previous studies, we have identified the cyclic AMP receptor protein (CRP) binding site, also known as the CAP site, in the promoter region upstream of the *carocin* gene [2,3]. Additionally, we have found that glucose can induce the expression of the carocin genes. However, the relationship between the CRP protein and the expression of the carocin protein is still not fully understood. Our objective in this study is to gain a deeper understanding of the signaling processes that lead to the production of carocin, a low-molecular-weight bacteriocin synthesized by *Pectobacterium carotovorum* subsp. *carotovorum* (*Pcc*). The *Pcc* virulence mechanism makes use of a variety of secretion systems (TSS) to export proteins or bacteriocins to the extracellular space. The Type III Secretion System (T3SS) is a large nanomachine used by symbiotic and pathogenic bacteria to deliver effector proteins directly into the cytoplasm of host cells, and its involvement in bacteriocin production is of particular interest in this study [14,15,16].

The cyclic AMP receptor protein (CRP), also referred to as the catabolite activator protein (CAP), has a crucial function in the modulation of gene expression in *E. coli* and other Gram-negative bacteria [17]. It interacts with a significant number of high-affinity sites, numbering over 300, and governs the activity of more than 500 genes in *E. coli* [18]. Moreover, CRP exhibits binding to an extensive set of low-affinity sites, surpassing 10,000, within the *E. coli* genome. This characteristic demonstrates its involvement not only as a specific transcriptional regulator but also as a protein that contributes to shaping the chromosome structure [19,20]. Genes involved in metabolism can be favorably or adversely regulated by CRP. Studies indicate that other molecules may be able to attach to CRP in addition to cAMP, which is normally the second messenger that does so. For instance, c-di-GMP, a physiological effector, but not adenylate cyclase or cAMP, can activate the cAMP receptor-like protein (CLP), a member of the CRP superfamily, in *Xanthomonas axonopodis* pv. *citri* [21]. In a different investigation, it was discovered that the cAMP-CRP complex in *V. cholerae* is related to the quantity of cyclic-di-GMP, which functions as a second messenger in the enzyme diguanylate cyclase (DGC) [22].

The cyclic AMP receptor protein (CRP) of bacteria that binds cAMP may function as a transcription factor that mediates activation associated with the synthesis of flagellum and the manufacture of toxin [23]. Interestingly, we have discovered that *Pcc* does not express bacteriocin in *dgc* or *crp* mutants.

CRPs have long been thought to play a role in metabolism and anti-oxidation pathways. In *E. coli* CRP, cAMP functions as a crucial signaling molecule and aids in the formation of the CRP-DNA complex [17], but no data can yet be displayed in *Pcc*. As a result, our study is the first to explore the possibility that the region upstream of the carocin gene contains potential *crp*-binding sites. It also looks at whether *crp* might interact with c-di-GMP to form a complex that might control how bacteriocin genes are expressed via T3ss. By investigating the mechanisms underlying carocin production, we hope to gain new insights into the regulation of bacterial virulence factors and pave the way for the development of new strategies to combat bacterial infections.

## 2. Results

### 2.1. Construction of CRP Mutant

Nucleotide sequence of *crp* obtained from Erwinia chrysanthemi [24] was utilized as a template for the development of a primer to isolate *crp* in *Pcc*’s genomic DNA. A 504 bp *crp* fragment was amplified through a polymerase chain reaction and was purified and cloned to a plasmid pBRH39 for sequencing. The nucleotide sequence and deduced amino acid of the putative protein were compared by the BLAST programs of the National Center for Biotechnology Information server (National Library of Medicine, Bethesda, MD, USA). Sequence data were compiled with DNASIS-Max 3.0 software (Hitachi, Chiyoda City, Japan). Results demonstrated that *crp* has a complete open reading frame of 632 bp and can be translated into 210 amino acids.

We discovered that Pectobacterium of the same genus shares the *crp* gene with the wild *Pcc* strain we chose for our laboratory (Figure 1).

The alignment of DNA and amino acid sequences of the *crp* genes showcased a striking level of similarity among different bacterial species. Specifically, the *crp* genes of the *Pcc* wild-type strain, Escherichia coli, Scandinavium, and Yersinia pestis demonstrated an impressive 99% similarity. This finding highlights the conserved nature of the *crp* gene across these bacterial strains and hints at its crucial role in their biological processes.

### 2.2. CRP Knockout Experiments

To validate the hypothesis that *crp* plays an important role in the expression of bacteriocin in *Pcc*, the multi low-molecular-weight bacteriocin producer H-rif-8-6 strain was utilized to block *crp* by homologous recombination.

The recombinant DNA fragment was electroporated into H-rif-8-6, and effective disruption of the low-molecular-weight bacteriocin-related genes was confirmed by bacteriocin tests [25,26] (Figure 2a). Each isolate’s inhibitory zone around the indicator strain Ea1068 was measured and contrasted with the inhibitory zone surrounding H-rif-8-6 upon transformation. Knockout strains 2 to 9 had no LMWB secretions, according to this evidence of mutation, with the exception of strain number 10, but the wild-type had normal LMWB secretion at the cellular level as observed from its larger zone of inhibition. The results from the modified Drigalski’s medium also showed that the wild-type H-rif-8-6 and CRP knockout mutants that were subjected to bacteriocin testing confirmed that the strains are *Pcc* (Figure 2b). Because strain 10 failed to exhibit a golden yellow colony appearance on the modified Drigalski’s medium, the test results excluded it from the criteria.

A transcriptional study was also carried out on how the bacteria produce, regulate, and secrete bacteriocin. To confirm if the *crp* gene has lost the function, both the wild-type and the *crp* knockout (*crp*-KO) strains were exponentially grown and induced to allow the cells to produce the bacteriocin. RNA was extracted from the harvested cells and subjected to in vitro analysis using a two-step end-point Reverse Transcriptase Polymerase Chain Reaction (end-point RT-PCR). The mRNA expression levels were then analyzed to determine the impact of the *crp* gene on the ability of the cells to produce, regulate, and secrete bacteriocin [4,5].

The agarose gel electrophoresis of Figure 3 revealed the absence of the *crp* gene band in the *crp*-KO mutant, indicating a successful knockout. Furthermore, Figure 3 makes it very evident that all of the genes under test have had their expression downregulated, suggesting that these are transcriptionally affected by *crp*.

### 2.3. Purification and Characterization of CRP

The CRP protein used in our study was derived from the full open reading frame region of *Pcc*’s CRP gene. The *crp* gene was subcloned into the pET32a vector, resulting in the expression of a CRP fusion protein with a His-tag at the N-terminus. This His-tag facilitates the purification of the CRP fusion protein and allows for the generation of polyclonal antibody serum. The pE-CRP-1 construct, which contains the *crp* gene within the pET32a vector, was used to express and produce the purified protein in *E. coli* BL21 (DE3) cells. Transformants were induced with IPTG under the control of the T7 promoter. The proteins were isolated and quantified by 40~50% ammonium sulfate precipitation followed by chromatographic separation of cell lysate.

The SDS-PAGE gels used to analyze purified CRP and stained with Coomassie blue, as described in Figure 4, revealed distinct lanes. Lane M contained the protein marker, with size markers (in kilodalton) indicated on the left side of the gel. The observed band in lane 2 corresponded to pET32a, a plasmid that exclusively produces a His-tag. This finding confirms the successful subcloning of the *crp* gene into the plasmid. The prominent band in lane 2 can be attributed to the overexpression of the protein following IPTG induction. In lane 3, the band represented the CRP-His fusion protein.

Figure 4a illustrates the isolated and purified full-length CRP protein, while Figure 4b demonstrates the CRP protein that underwent purification via a nickel column. For the purification process, the CRP band (indicated by an arrowhead in Figure 4a) was isolated by eluting the column using a salt-gradient method. Gel analysis in Figure 4b confirmed the presence of a distinct band representing CRP with an approximate molecular weight (Mr) of approximately 41.3 kDa. Notably, other bands present in the gel were no longer visible, indicating the successful purification of CRP.

### 2.4. Production of Anti-CRP Polyclonal Antibody

In order to generate polyclonal antibodies, an immunization protocol was performed using BALB/c mice. The purified CRP fusion protein was administered to five adult female mice via subcutaneous injections. The immunization process involved a total of four injections, with a two-week interval between each injection, as shown in Figure 5a. Blood samples were collected at different time points starting from the beginning of immunization. These blood samples were subsequently analyzed using ELISA assays to evaluate the progress of the specific immune response, as depicted in Figure 5b.

A solid-phase ELISA test of the mice sera (from several serum dilution) prepared from these samples revealed an increase in reactivity toward CRP after six weeks of immunization (Figure 5b).

### 2.5. Specificity of CRP

The ability of the mouse polyclonal CRP antibodies to bind with CRP monomers and dimers suggests that they were selective.

To investigate the impact of CRP on the production of low-molecular-weight bacteriocins, we examined the production of carocin in various *Pcc* strains, including H-rif-8-6 (wild-type), H-rif/Δ*crp* (*crp* knockout strain), Rif-TO6 (wild-type), and TT6-6 (*dgc* knockout strain). We conducted Western blot analysis on cell lysates containing native CRP protein and compared them to lysates without CRP protein to confirm the specificity of the CRP antibody serum. The cell lysates from the wild-type strains exhibited specificity, as evidenced by the presence of a distinct band at approximately 43.1 kDa molecular weight, as observed in the 12% SDS-PAGE and Western blot examinations (Figure 6c,d).

Moreover, our focus was on the multi-bacteriocin producer *Pcc* strains H-rif/Δ*crp* (*crp* knockout), Rif-TO6 (wild-type), and TT6-6 (dgc knockout strain) since our main objective was to assess the control of CRP in *Pcc* strains on the low-molecular-weight bacteriocin, carocin. To compare the ability of the CRP antibody serum to bind to different cell lysates containing or lacking native CRP protein, we performed a Western blot analysis following a 12% SDS-PAGE (Figure 6c,d). The cell lysates from the wild-type strains demonstrated specificity in both the SDS-PAGE and Western blot analyses, indicated by the presence of a color band at approximately 43.1 kDa molecular weight. 

In contrast, we observed that the *crp* knockout strain (H-rif/Δ*crp*) did not show any color band for the CRP protein in either test, while the wild-type counterpart (TO6) and the mutant carrying the *dgc* knockout (TT6-6) displayed the color band for the CRP protein. These results indicate that the expression of CRP is unaffected by the *dgc* gene mutation.

### 2.6. Analysis of CRP and Promoter Binding Region

According to reports, CRP regulates virulence factors in the bacteria *Yersinia pestis*, *Escherichia coli*, and *Salmonella enterica* [27]. It is also evident that the CRP protein is linked to the genes for bacteriocin in both *Yersinia pestis* and *Escherichia coli*. As *Pcc*’s *crp* and these strains share homology, we aimed to investigate the CRP-binding region on carocin to better understand the regulatory functions of CRP.

Analysis of the DNA sequence upstream of the translation initiation site of carocin S3 using DNASIS-Max 3.0 software (Hitachi, Chiyoda City, Japan) revealed two putative binding sites for CRP, which we named CAP site 1 and CAP site 2 (Figure 7). These sites are located at −112–−117 bp and −17–−22 bp upstream of the translation start point, respectively. To confirm these findings, we conducted biotinylated probe precipitation experiments, which conclusively showed that CAP sites 1 and 2 are actually bound to CRP.

To identify CRP binding to biotinylated CAP sites in the presence or absence of UV stimulation, we conducted a Western blot analysis using the CRP antibody serum that we generated. In the wild-type strain H-rif-8-6, which was not induced by UV, we observed that the CRP protein binds more strongly to CAP site 1 than to CAP site 2. However, when this strain was exposed to UV stimulation, the affinities between CRP and both CAP site 1 and CAP site 2 improved and became more comparable (Figure 8a). In contrast, in the Rif-TO6 strain without UV stimulation, we observed that the binding capacity of the CRP protein to CAP site 2 was higher than that of CAP site 1. However, after UV stimulation, the affinity between CRP and CAP site 1 increased.

The dgc knockout strain TT6-6 lacks the ability to cyclize two molecules of GTP into c-di-GMP. We observed that, regardless of UV exposure, the TT6-6 mutant strain showed significantly lower binding capacity to CAP site 1, CAP site 2, and CRP protein than the Rif-TO6 wild-type strain (Figure 8b). This observation might be caused by the TT6-6 mutant strain lacking the secondary messenger c-di-GMP.

These observations support the hypothesis that UV absorption by DNA-associated proteins may cause DNA damage and potentially impact bacterial gene expression, although this aspect was not specifically investigated in this study.

## 3. Discussion

In this study, we provide evidence for the regulatory role of the *crp* gene in controlling the expression of bacteriocins in *Pectobacterium carotovorum* subsp. *carotovorum* H-rif-8-6. The complete open reading frame of CRP consists of 632 base pairs and encodes a protein of 210 amino acids, as determined through a BLAST search. Our analysis revealed similarities between the *crp* genes of *Yersinia pestis*, *Pectobacterium wasabiea*, and *Salmonella enterica*. Previous studies have shown that CRP influences the virulence factors in *Yersinia pestis*, *Escherichia coli*, and *Salmonella enterica* [28], and it is also associated with bacteriocin genes. While the forward promoter (promoter) controls the transcriptional activity of bacteriocin genes [29], the specific regulatory link between CRP proteins and virulence genes in Pectobacterium has not been elucidated.

We found that the *crp* gene is essential for the production of the low molecular weight bacteriocins caroS2K, caroS3K, and caroS4K, as well as genes related to the T3bSS system. These results suggest that CRP may play a significant role in the regulation of bacteriocin synthesis, either as an upstream regulator or even as a master switch of the entire regulatory system.

Previous studies have shown that bacterial cells often produce bacteriocins in response to stressful conditions. For example, in *E. coli*, the production of colicin is induced by the DNA damage repair system known as the SOS system [30], which is controlled by the LexA (repressor) and RecA (inducer) proteins. Some research suggests that RecA can activate the formation of carotovoricin [31], which has been linked to the SOS system in *Pcc*. Cyclic-di-GMP and cyclic-AMP, both of which are connected to the regulatory genes of the SOS system [32] and have been shown to interact with CRP. These findings further support our hypothesis regarding the regulation of *Pcc* bacteriocins.

Although the SOS boxes are not present upstream of the carocin genes, our analysis using DNASIS Max 3.0 software identified two binding sites known as CAP site 1 and CAP site 2. These binding sites, designated as CIE-1 and CIE-2 (carocin induced element-1 and carocin induced element-2), are located upstream of the carocin genes. Understanding the CRP-binding site is essential to determine the regulatory functions of CRP. The cAMP-CRP complex can have opposing regulatory effects, activation, or repression when interacting with a binding site for gene regulation [33].

Generally, activation of CRP requires binding with cAMP [34]. According to several studies on the structure of the CRP protein, CRP dimers can form complexes such as CRP-cAMP, CRP-cAMP2, CRP-cAMP3, or CRP-cAMP4, depending on the concentration of cAMP available.

Surprisingly, our biotinylated probe precipitation study revealed that the binding capacity of CRP to CAP site 1 and CAP site 2 of the TT6-6 strain was significantly lower in the absence of dgc compared to the Rif-TO6 wild-type strain. This observation suggests that c-di-GMP, in addition to cAMP, may act as a regulatory molecule. When bound to the CRP protein, c-di-GMP might induce structural changes in CRP and increase its affinity for CAP sites 1 and 2. Previous reports have indicated that the interaction of two molecules of cAMP and CRP dimer can promote CRP isoform conversion [35], leading to the positive regulation of gene expression by the resulting CRP-cAMP2 complex.

Based on our assumptions, it is likely that the cAMP-CRP complex can directly regulate the *dgc*, *brg*, *flhA*, and *glnH* genes by binding to their regulatory sites [29]. Furthermore, CRP may indirectly regulate genes involved in the flagellar system through DGC. DGC is responsible for the production of c-di-GMP [22], which is a secondary messenger associated with bacterial virulence. We previously hypothesized that CRP might be influenced by interaction with c-di-GMP due to the lack of information on the adenylyl cyclase (AC) and cAMP receptor-like protein (CLP) sequences in *Pcc*.

In the recent update of the NCBI database, we identified the AC gene cyaB and the cAMP receptor-like protein (clp) through BLAST analysis, confirming their presence in *Pcc*. Our findings indicate the existence of four different types of *clp* genes in *Pcc*. Additionally, we observed the presence of the PilZ domain-containing flagellar brake protein YcgR, which serves as a second receptor for c-di-GMP and can regulate the *fliG* gene [36,37].

The synthesis of bacteriocins is regulated by multiple regulatory systems, and one of the key regulators is the CRP-cAMP complex. We propose that the CRP-cAMP complex directly controls the T3bSS system, which includes the *flhA* and *brg* genes, as well as the ABC transporters represented by *glnH* (Figure 9). Our hypothesis suggests that the CRP-cAMP complex binds to the regulatory regions of these genes and modulates their transcription, thereby influencing bacteriocin synthesis.

Knocking out the *crp* gene impairs the functioning of these systems, leading to the suppression or induction of bacteriocin secretion. In addition to direct regulation, there are indirect mechanisms involving the *dgc* and *cyaB* genes that are activated by the CRP-cAMP complex, promoting the synthesis of c-di-GMP. We propose that the YcgR protein contains two c-di-GMP receptors: the CLP and PilZ domains. These domains regulate the *fliC* and *fliZ* genes, respectively, which enables the T3bSS to release bacteriocins.

## 4. Materials and Methods

*Bacterial strains, plasmids, media, and growth conditions.* Table 1 and Appendix A provide a list of the bacterial strains, plasmids, and primers used in this study. On a modified Luria–Bertani (LB) medium containing 5 g of sodium chloride per liter at 28 °C, *Pcc* strains were grown (half from the recommended quantity of NaCl). *E. coli* (the cloning host) strains were grown in LB broth at a temperature of 37 °C and 125 rpm. Then, 1% polypeptin, 0.2% yeast extract, 0.1% MgSO_4_ (pH 7.0), and 1.5% agar were added to the IFO-802 medium. *E. coli* and *Pcc* strains were treated with antibiotic doses used in bacterial selection: 50 µg/mL rifampicin, 50 µg/mL kanamycin, and 50 µg/mL ampicillin. Using a spectrophotometer, all bacterial growth densities were calculated at 595 nm (OD_595_).

*Bacteriocin assays.* The soft agar overlay method was used to evaluate the bacteriocin-producing strain’s antibacterial activity [40]. The isolates were grown on hard IFO-802 (1.4% agar) and soft IFO-802 (0.65% agar) media. The cells were first incubated for 12 h before colonies could form. The colonies were then irradiated in an open glass Petri dish at a distance of 40 cm from a 40 Watt UVC/254 nm germicidal lamp (Sekyo-Denki, Japan) and cultured for an additional 12 h. Before the soft agar with the indicator cells was applied, the cells were treated with chloroform. An inhibiting zone of indicator-cell (EA1068) growth surrounding the colony serves as a sign that bacteriocin is being produced.

*Preparation of genomic DNA, plasmid DNA, and RNA.* Standard procedures were followed for DNA ligation, agarose gel electrophoresis, DNA purification from agarose gels, and other cloning-related techniques [41,42]. For RNA extraction, exponentially developing *E. coli* DH5α cells (OD_595_ of approximately 1.0) were taken out. According to the manufacturer’s instructions, RNA was extracted using Trizol reagent (Invitrogen, Massachusetts, USA) and resuspended in DEPC-treated water. Utilizing a NanoVue Plus^TM^ spectrophotometer (Biochrom, Massachusetts, USA), the purity and concentration of total RNA were determined. Electrophoresis on a 1.5% formaldehyde–morpholinepropanesulfonic–agarose gel was then performed for evaluation. 

*Bacterial matting.* According to Gantotti et al. [43], bacterial mating was accomplished using the membrane-filter approach. *Pcc* (receiver) H-rif-8-6 and *E. coli* (donor) 1830 cultures were incubated overnight at 28 °C on LB agar media and evenly distributed on 0.22 µm pore size membrane filters (Millipore, Inc., Bedford, MA). Following conjugation, progeny suspensions were appropriately diluted and cultured at 28 °C for 24 to 48 h on modified Drigalski’s agar plates (containing rifampicin and kanamycin, 100 µg mL^−1^). Colonies were isolated for the test for bacteriocin production.

*Preparation of CRP protein.* The purification of cAMP receptor protein (CRP) from an *E. coli* BL21 expression strain transformed with pE-CRP-1 plasmid involves several steps. Initially, a test tube containing 1 mL of LB broth supplemented with antibiotics is prepared, and a single bacterial colony is selected and inoculated. The culture is incubated overnight at 37 °C with shaking. The following day, 500 μL of the overnight culture is transferred to a flask containing 50 mL of fresh LB broth, resulting in an amplified culture with an LB-to-bacterial-culture ratio of 100:1. The culture is shaken at 37 °C until the optical density at 595 nm (OD_595_) reaches approximately 0.4–0.5. At this point, 6 μL of 0.8 M IPTG is added to induce protein expression, and the culture is further incubated at 28 °C for 5 h.

After the induction period, the bacterial culture is poured into a centrifuge bottle and centrifuged at 4 °C/6000 rpm for 20 min to collect the bacterial cells. The cell pellet is then resuspended in 10 mL of loading buffer. The suspended cells are transferred to a container for bacterial lysis, such as a 50 mL centrifuge tube with a cut-off conical bottom. To maintain a low temperature, the container is placed in a beaker filled with ice cubes and secured with rubber bands. An ultrasonic cell smasher with a power setting of 200 W and a pulse/pause ratio of 3/5 s is used to disrupt the bacterial cells. Care is taken to avoid excessive disturbance of the liquid level during this process, as well as to prevent high temperatures that may lead to protein denaturation. Once the suspension turns from turbid to clear, the lysed mixture is centrifuged at 4 °C/10,000 rpm for 10 min to remove unbroken cells.

*Characterization of CRP protein.* To characterize cAMP receptor protein (CRP), sodium dodecyl sulfate polyacrylamide gel electrophoresis (SDS-PAGE) analysis is performed. For gel preparation, two plastic glass pieces are assembled with a spacer and fixed with clips. The lower gel is prepared and poured between the glass slides, followed by the upper gel with the toothed comb inserted. After gel solidification, the comb is removed. In the electrophoresis tank, 1X TG-SDS buffer is added, and marker and samples are injected into the sample tank. Electrophoresis is conducted at 80 volts for 20 min and then at 110 volts until the tracking dye reaches the gel bottom.

For colloid staining, the gel is carefully removed and placed in a fresh-keeping box containing Coomassie Brilliant Blue R-250 (CBR) staining solution. After shaking for 30 min, excess CBR solution is poured off, and destaining solution is added. Shaking for another 30 min removes the background, revealing clear bands of stained colloid.

Protein quantification is performed using the Bradford method. The protein solution is mixed with the appropriate Bradford reagent, and after a 2 min incubation, the absorbance at OD_595_ is measured. A standard calibration line is generated using known concentrations of BSA, and the OD_595_ of the protein solution is measured to determine its concentration by referring to the calibration line equation.

*Immunization and serum harvest.* Next, 5 female mice were immunized subcutaneously with 200 μL of antigen solution containing approximately 200 μg of *crp*, using either Freund’s complete adjuvant for the initial immunization or Freund’s incomplete adjuvant for subsequent booster immunizations at 2-week intervals. After titers had reached acceptable levels, 1 mouse was chosen for antibody production and maintained for production bleeds (a 6-week cycle of immunization). All blood samples were refrigerated for 24 h and centrifuged (450× *g*; 10 min) at room temperature. The serum was then stored at −20 °C.

## 5. Conclusions

*Pcc* produces a bacteriocin that is secreted extracellularly as a hybrid of the injectisome and flagellar secretion systems that can kill its gene-related strains in response to environmental stimuli such as nutritional deficiency or UV exposure. The cyclic AMP receptor protein (CRP), which is involved in the extracellular export of bacteriocins via the flagellar type three secretion system, affects the expression of various low-molecular-weight bacteriocins and suppresses the expression of genes related to this process. Our findings suggest that under UV induction, CRP preferentially binds to both of the CAP sites, while in the absence of UV induction, it binds to only one. This new understanding of how bacteriocin synthesis and secretion are controlled in *Pcc* may open up the possibility to the development of novel antibacterial approaches.

## Figures and Tables

**Figure 1 ijms-24-09752-f001:**
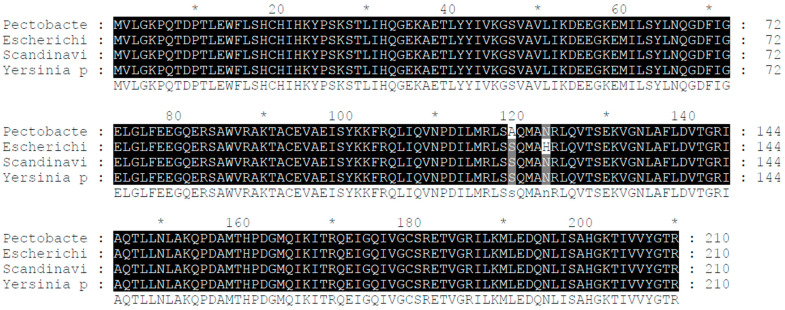
Alignment of the *crp*’s DNA and amino acid sequence. The *crp* gene of the *Pcc* wild-type strain shows a remarkable similarity of 99% with the *crp* genes of *Yersinia pestis* and *Escherichia coli*, indicating a high degree of conservation.

**Figure 2 ijms-24-09752-f002:**
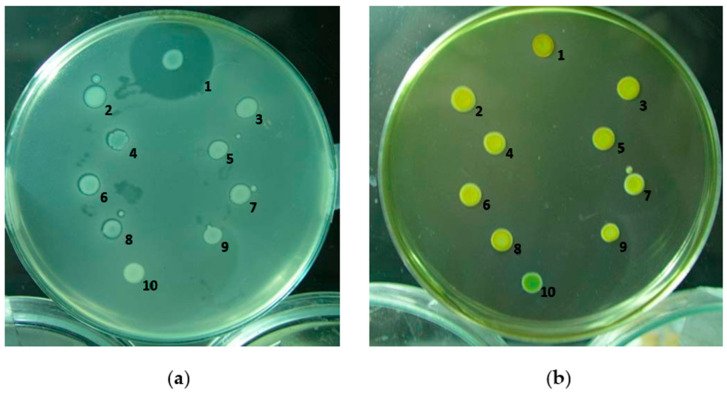
Bacteriocin assay of the gene knockout strain. (**a**) Bacteriocin activity of the gene selection strain. The zone of inhibition against the indicator strain is Ea1068 and indicates the expression of the low-molecular-weight bacteriocins. 1: H-rif-8-6; 2: RH1; 3: RH2; 4: RH3; 5: RH4; 6: RH5; 7: RH6; 8: RH7; 9: RH8; 10: RH9. (**b**) Screening of mutants by modified Drigalski’s medium. If the strain is *Pcc*, the medium will be golden yellow, but if it is an *Escherichia coli* colony, the medium will be blue. 1: H-rif-8-6; 2: RH1; 3: RH2; 4: RH3; 5: RH4; 6: RH5; 7: RH6; 8: RH7; 9: RH8; 10: RH9 (RH is a code for crp knockout strain).

**Figure 3 ijms-24-09752-f003:**
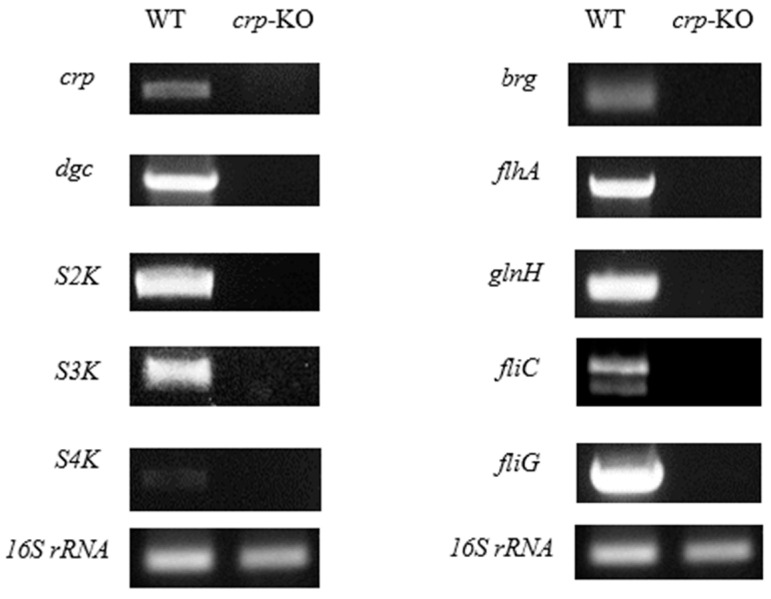
Transcriptional analysis of bacteriocin genes and regulatory genes by two-step end-point RT-PCR. Gel electropherogram of RT-PCR products exhibited by the wild-type and *crp* defective mutants are shown.

**Figure 4 ijms-24-09752-f004:**
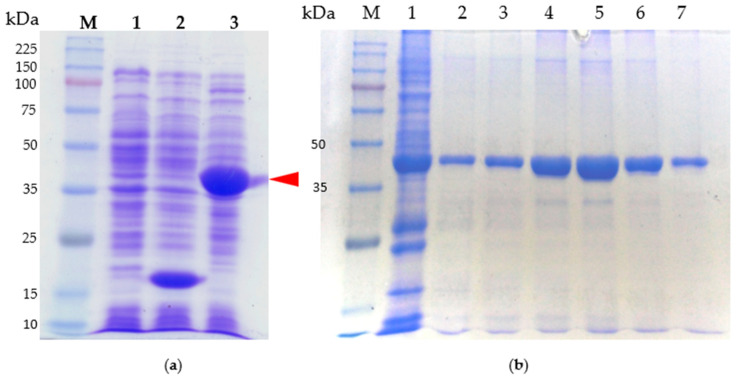
SDS-PAGE analysis of purified protein. Shown are the CRP protein: (**a**) Samples were subjected to electrophoresis in 10% polyacrylamide gels, which were stained with Coomassie blue. Lane M, molecular weight standards (kDa); lane 1, cell lysate of *E. coli* BL21; lane 2, cell lysate of BL21/pET32a; lane 3, cell lysate of BL21/pE-CRP-1. The arrowhead indicates the overexpression of CRP fusion protein. (**b**) CRP fusion protein purification process. Lane M, molecular weight standards (kDa); lane 1, cell lysate of BL21/pET32a; lanes 2–7 purified protein after elution.

**Figure 5 ijms-24-09752-f005:**
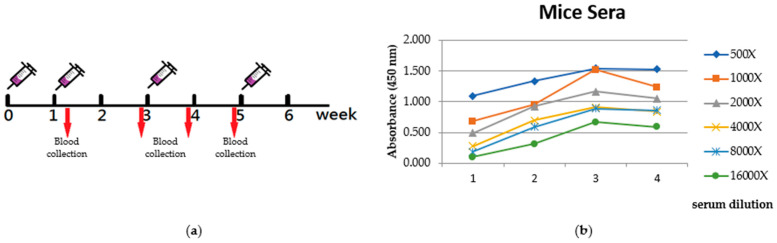
Evaluation of immune response. (**a**) Time course of immunization. (**b**) Reactivity of blood sera from CRP immunized mice ELISA test. Serum samples (diluted at different concentrations) were taken from the immunized mice at several time points (days) from the first injection onwards. Antibody/antigen interaction was detected by goat anti-mice HRP conjugated antibody.

**Figure 6 ijms-24-09752-f006:**
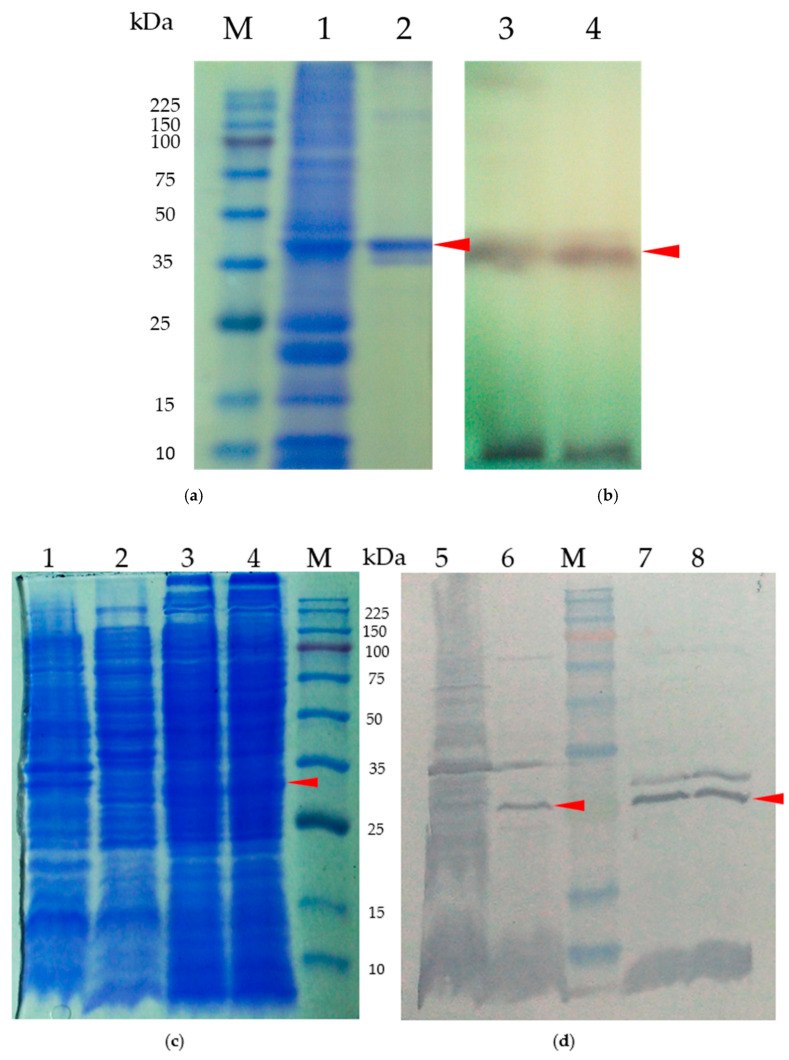
(**a**) Test of the CRP polyclonal antibody serum’s specificity for the CRP fusion protein. Samples were subjected to electrophoresis in 10% polyacrylamide gels, which were stained with Coomassie blue. Lane M, molecular weight standards (kDa); lane 1 represents the cell lysate and lane 2 represents the purified CRP protein. (**b**) Anti-CRP Western blot analysis using cell lysates from pE-CRP-1/BL21 purified product. Lane 3 corresponds to the cell lysate, while lane 4 represents the purified CRP protein. (**c**) SDS-PAGE of purified native protein for CRP polyclonal antibody serum specificity testing. (**d**) Cell lysates from the *crp* mutant strain (*Pcc* H-rif-8-6/Δ*crp*) and wild-type strain; lanes 1 and 5 represent *crp* mutant strain (*Pcc* H-rif-8-6/Δ*crp*, lanes 2 and 6 represent the wild-type strain (Rif-TO6), lanes 3, 4, 7, and 8 represent the complementary experiment using the *dgc* knockout strain (TT6-6) and were studied by Western blotting using an anti-*crp* antibody. The arrows indicate the color bands of CRP proteins.

**Figure 7 ijms-24-09752-f007:**
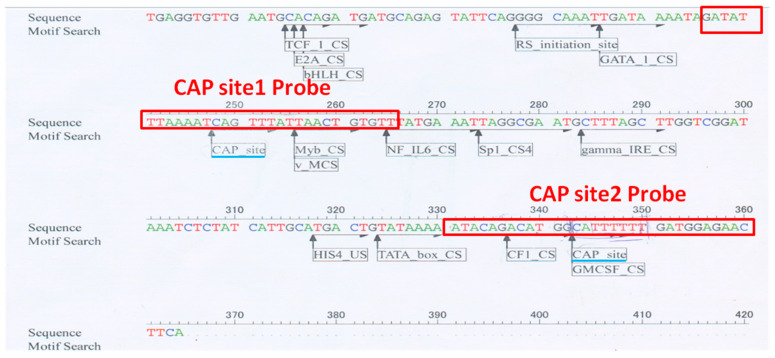
The sequence of the carocin S3 gene’s upstream non-translated region was examined using DNASIS Max 3.0 software. The 30 bp DNA fragment was employed as a probe, and the analyzed CAP site location indicated that it extended 12 bp upstream and downstream; the light blue bottom line represented the studied CAP site position by the DNAIS MAX program. The biotin-labeled probe’s DNA fragment is shown in the red box.

**Figure 8 ijms-24-09752-f008:**
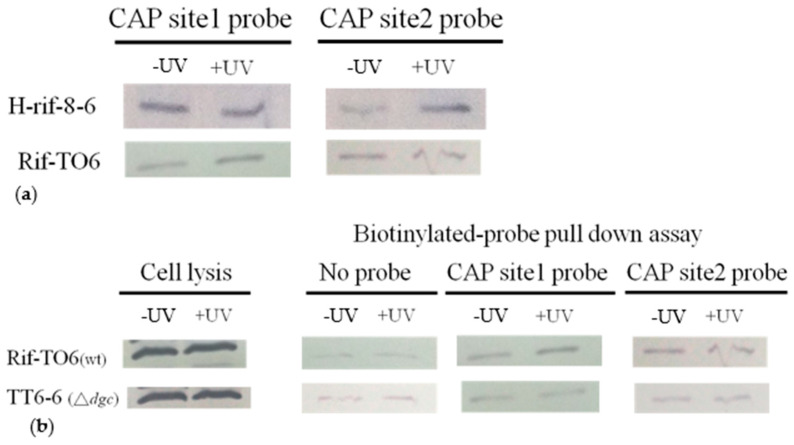
Representative immunoblots displaying the simultaneous detection of the *crp* that bind to the biotinylated CAP sites in the presence or absence of UV induction. (**a**) Comparison of the two wild-type strains’ *crp* proteins’ capacity to bind to *crp* CAP sites. (**b**) Comparison of the ability of *crp* proteins from wild-type and dgc knockout strains to bind to *crp* CAP sites. Bacterial strains that were stimulated by UV for 90 s and then cultured in the dark for 30 min were referred to as +UV; those that were not stimulated by UV for bacteriocin expression were referred to as −UV.

**Figure 9 ijms-24-09752-f009:**
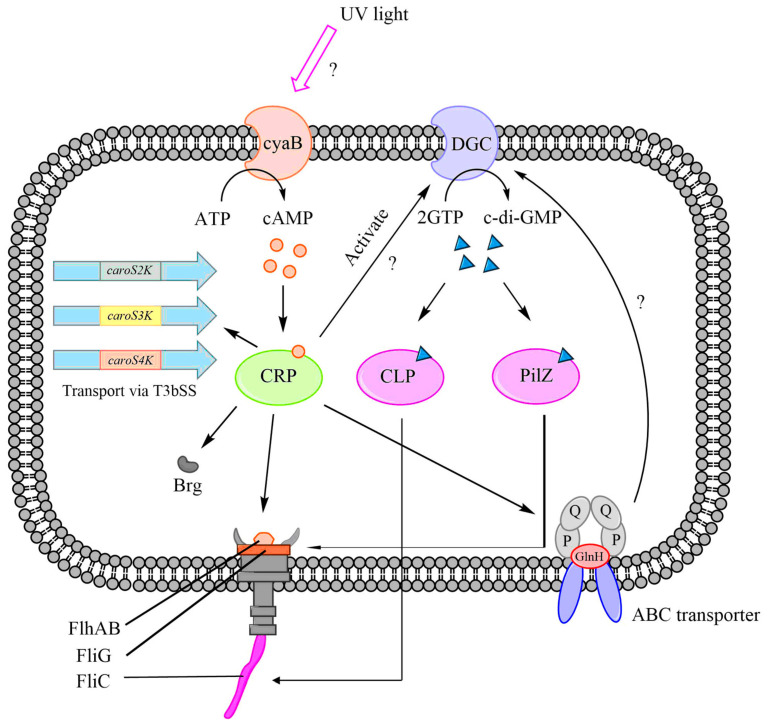
The bacteriocin regulation mechanism with the CRP-cAMP complex. A hypothesized role for *crp* in the production of carocins and their secretion through T3bSS.

**Table 1 ijms-24-09752-t001:** Bacteria and plasmids used in this study.

Strain or Plasmid	Relevant Characteristics	Source of Reference
** *Escherichia coli* **		
DH5α	SupE44 ΔlacU169 (Φ80 lacZ ΔM15) hsdR17 recA1 endA1 gyrA96 thi-l relA1	Hanahan; Reusch et al. [38,39]
BL21(DE3)	hsdS gal(λcIts857 ind1 Sam7 nin5 lacUV5-T7 gene 1)	Novagen
***Pectobacterium carotovorum* subsp. *carotovorum***		
TO6	*Pcc*, Amp^r^, wild-type, putative biocontrol agent	Laboratory stock
Rif-TO6	*Pcc*, Amp^r^, Rif^r^, wild-type	Laboratory stock
89-H-4	*Pcc*, Amp^r^, wild-type, putative biocontrol agent	Laboratory stock
89-H-rif-12	*Pcc*, Amp^r^, Rif^r^, wild-type	Laboratory stock
3F3	*Pcc*, Amp^r^, wild-type	Laboratory stock
EA1068	*Pcc*, wild-type indicator strain	Laboratory stock
H-rif-CK (Δ*crp*::*kan*)	H-89-4, Δ*crp*, Rif^r^, Kan^r^	This study
H-rif-8-6	*Pcc*, Amp^r^, wild-type, putative biocontrol agent	This study
TT6-6	TO6, Δ*dgc*, Rif^r^, Kan^r^	This study
** *Plasmids* **		
pGEM T-Easy	Amp^r^, lacZ cloning vector	Promega
pBR322	Amp^r^, Tet^r^	Bolivar et al. [26]
pG-*crp*	pGEM T-Easy Vector, *crp* gene Amp^r^	This study
pG-CT (Δ*crp*::*tet*)	pGEM T-Easy Vector, *crp* gene Amp^r^, Tet^r^	This study
pG-Tet	pGEM T-Easy, Amp^r^, Tet^r^	Laboratory stock

## Data Availability

Not applicable.

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
