# Peer review of "Unleashing the Influence of cAMP Receptor Protein: The Master Switch of Bacteriocin Export in Pectobacterium carotovorum subsp. carotovorum"

_ijms, 2023, doi:10.3390/ijms24119752_

Round 1
Reviewer 1 Report
The manuscript entitled "Unleashing the influence of cAMP Receptor Protein: The master switch of bacteriocin export in Pectobacterium carotovorum subsp. caratovorum" found it to be an interesting study. the author-generated data related to CRP is novel. I would expect the monoclonal anti-body generation of CRP and its epitope identification in E. Coli. the author should show such data to improve it.
The manuscript entitled "Unleashing the influence of cAMP Receptor Protein: The master switch of bacteriocin export in Pectobacterium carotovorum subsp. caratovorum" found it to be an interesting study. the author-generated data related to CRP is novel. I would expect the monoclonal anti-body generation of CRP and its epitope identification in E. Coli. the author should show such data to improve it.
Author Response
May 18, 2023
The Reviewer,
International Journal of Molecular Sciences
Madam/Sir,
In the article " Unleashing the influence of cAMP Receptor Protein: The master switch of bacteriocin export in Pectobacterium carotovorum subsp. caratovorum" I'd like to express my gratitude for taking the time to review the aforementioned paper. Indeed, your suggestions are admirable, giving us optimism that our manuscript may be accepted in this in this prestigious publication.
In view of this, the following is a point-by-point response:
- I would expect the monoclonal anti-body generation of CRP and its epitope identification in E. Coli. the author should show such data to improve it.
- The different sections in the article were already revised based on the recommendations of the reviewers. The generation of monoclonal anti-body generation of CRP can not be obtained at present but may also be considered in the future.
Please find attached corrected manuscript for your reference.
Thank you and more power!
Yours Truly,
Duen-Yau Chuang, PhD
Corresponding Author
Reviewer 2 Report
Dear authors,
This is an original work submitted by Chang et al. entitled: Unleashing the influence of cAMP Receptor Protein: The master switch of bacteriocin export in Pectobacterium carotovorum subsp. Caratovorum.
This manuscript is well-constructed, presenting enough results on the bacteriocin synthesis and secretion controlled in Pcc. However, I have some comments and suggestions to be addressed.
Major revisions are suggested before the publication of the manuscript.
- Lines 10 and 11 I am not sure about the politics of the journal on the authors' contact information but in my opinion, e-mail must be institutional one.
- Beware of similarities between your paper and others published previously. For example, lines 57 to 61 of your manuscript are almost the same as those published by Leinisch, et. al. (Reference 18 of your work).
Your work: “The cyclic AMP receptor protein (CRP), also known as the catabolite activator protein (CAP), plays a major role 57 in regulating gene expression in E. coli and other Gram-negative bacteria [18]. CRP binds to over 300 high-affinity 58 sites and controls more than 500 genes in E. coli [19]. Additionally, CRP binds to more than 10,000 low-affinity sites 59 in the E coli genome, demonstrating that it is a chromosome-shaping protein in addition to being a particular tran-60 scriptional regulator [20,21].”
Leinisch, et. al.: “The cyclic AMP receptor protein (CRP) (also known as the catabolite activator protein, CAP) is a global regulator of gene expression in E. coli and other Gram negative bacteria. CRP regulates more than 500 genes in E. coli by binding to approximately 300 high-affinity sites 1. CRP also binds to more than 10000 low-affinity sites in the E. coli genome, indicating that CRP is not only a specific transcriptional regulator, but also a chromosome shaping protein, 2,3.”
- Experimental methodology for PCR Purification and Characterization was missing in Section 4.
- Homogenize Fig. and Figure, e.g. "Fig." was used on lines 175, 192 and "Figure" was used on lines 106, 133, 139, 176, 178, etc.
- Line 233 Change to "(Figure 5c and 5d)".
- Line 239 Add parentheses and change to "(Figure 6c and 6d)."
- Reference [24] does not appear in the text.
- Figures 6a and 6b appear to be on different scales although they are complementary experiments.
- Line 339 You miss a reference in your earlier proposal "PCR can be triggered through interaction with c-di-GMP".
- Bacteriosin assays are relevant to your study, especially UV radiation in your cultures, however, you do not specify the wavelength of UV light used in your experiments.
- In general, your discussion is like Brainstorming, please try to integrate all ideas to give fluidity to this section.
- Use the standard format for references, e.g. the DOI is missing for references 1 to 6. Sometimes you have used the full DOI URL (line 441) and only the DOI in other cases (lines 445 and 447), etc.
- Experimental methodology for Purification and Characterization of CRP was missed of Section 4.
In order to improve your work, the manuscript needs to be edited by a strong editor fluent in English.
Author Response
May 18, 2023
The Reviewer,
International Journal of Molecular Sciences
Madam/Sir,
In the article " Unleashing the influence of cAMP Receptor Protein: The master switch of bacteriocin export in Pectobacterium carotovorum subsp. caratovorum" I'd like to express my gratitude for taking the time to review the aforementioned paper. Indeed, your suggestions are admirable, giving us optimism that our manuscript may be accepted in this in this prestigious publication.
In view of this, the following is a point-by-point response:
- Lines 10 and 11 I am not sure about the politics of the journal on the authors' contact information but in my opinion, e-mail must be institutional one.
- The authors no longer have an active institutional e-mail since they already graduated in the laboratory.
- Beware of similarities between your paper and others published previously. For example, lines 57 to 61 of your manuscript are almost the same as those published by Leinisch, et. al. (Reference 18 of your work).
- The recommendation rewrite this portion was followed.
- Experimental methodology for PCR Purification and Characterization was missing in Section 4.
- PCR Purification and Characterization was added in the experimental methodology.
- Homogenize Fig. and Figure, e.g. "Fig." was used on lines 175, 192 and "Figure" was used on lines 106, 133, 139, 176, 178, etc.
- “Figure” was used all throughout the section.
- Line 233 Change to "(Figure 5c and 5d)".
- Changed to "(Figure 5c and 5d)
- Line 239 Add parentheses and change to "(Figure 6c and 6d)."
- parentheses was added.
- Reference [24] does not appear in the text.
-Reference 24 was added to the text.
- Figures 6a and 6b appear to be on different scales although they are complementary experiments.
- Figure 6a and 6b was fixed.
- Line 339 You miss a reference in your earlier proposal "PCR can be triggered through interaction with c-di-GMP".
- The statement was restated.
- Bacteriosin assays are relevant to your study, especially UV radiation in your cultures, however, you do not specify the wavelength of UV light used in your experiments.
- UVC with its corresponding wavelength was stated in the discussion.
- In general, your discussion is like Brainstorming, please try to integrate all ideas to give fluidity to this section.
- The whole discussion section was revised to give fluidity to this section.
- Use the standard format for references, e.g. the DOI is missing for references 1 to 6. Sometimes you have used the full DOI URL (line 441) and only the DOI in other cases (lines 445 and 447), etc.
- Standard format for references was followed.
- Experimental methodology for Purification and Characterization of CRP was missed of Section 4.
- PCR Purification and Characterization was added in the experimental methodology.
Please find attached corrected manuscript for your reference.
Thank you and more power!
Yours Truly,
Duen-Yau Chuang, PhD
Corresponding Author
Reviewer 3 Report
This study focused on gram-negative bacterium, Pcc and investigated the molecular mechanism of crp regulation on bacteriocin and export. By using gene knockout method and biotinylated probe pull-down experiment, the authors revealed the deletion of crp inhibited the genes in bacteriocin export mainly through T3SS system and impacted many bacteriocins’ production. Furthermore, it was found that CRP prefers to bind with two binding sites induced by UV, which is different from the one-site binding case when UV is missing. The followings are some concerns and suggestions for the authors’ consideration:
- Line 34: The citation in the first paragraph should be [1] or both [1] and [2].
- Experimental procedures are not provided or clearly described, for example, how was the gene knockout performed? There is not too much information about the DNA plasmid construction of crp mutant (primers, restriction enzymes, etc.).
- Line 418: As this study involved the use of animals, should it include any institutional review board statement?
- Line 87-101: Figure 1 is of low quality, too blurry. There is no real gene sequence alignment except for just the statement of the alignment results, which can be removed from the figure and put into the paragraph. Just keep the alignment of amino acid sequences.
- Line 116-130: In Figure 2, all number labels are too small, looks very blurry. And also, what are RH1-9? They should be clearly indicated for better understanding.
- Line 170-174: In Figure 4 legend, the lane 2 in (a) is the same as the lane 1 in (b), but the SDS-PAGE gels show different bands. Please clearly describe/explain the difference. What is the big band in the lane 2 in (a)? And also, delete “B” in “(b) B”
- Line 366 & Line 393: In the Materials and Methods section, is it true that antibiotic doses are used in the concentration of 50 g/ml? This is a very high concentration. And also, rifampicin and kanamycin as 100 g ml-1 in Bacterial Matting section?
- Line 425-427 & 439-441: Reference No.1 is the same as No. 7.
- Line 69: the citation is missing, or change to “discovered in this study that …”
- What is the pE-CRP-1 construct? It is not mentioned in the Materials and Methods.
This study focused on gram-negative bacterium, Pcc and investigated the molecular mechanism of crp regulation on bacteriocin and export. By using gene knockout method and biotinylated probe pull-down experiment, the authors revealed the deletion of crp inhibited the genes in bacteriocin export mainly through T3SS system and impacted many bacteriocins’ production. Furthermore, it was found that CRP prefers to bind with two binding sites induced by UV, which is different from the one-site binding case when UV is missing. The followings are some concerns and suggestions for the authors’ consideration:
- Line 34: The citation in the first paragraph should be [1] or both [1] and [2].
- Experimental procedures are not provided or clearly described, for example, how was the gene knockout performed? There is not too much information about the DNA plasmid construction of crp mutant (primers, restriction enzymes, etc.).
- Line 418: As this study involved the use of animals, should it include any institutional review board statement?
- Line 87-101: Figure 1 is of low quality, too blurry. There is no real gene sequence alignment except for just the statement of the alignment results, which can be removed from the figure and put into the paragraph. Just keep the alignment of amino acid sequences.
- Line 116-130: In Figure 2, all number labels are too small, looks very blurry. And also, what are RH1-9? They should be clearly indicated for better understanding.
- Line 170-174: In Figure 4 legend, the lane 2 in (a) is the same as the lane 1 in (b), but the SDS-PAGE gels show different bands. Please clearly describe/explain the difference. What is the big band in the lane 2 in (a)? And also, delete “B” in “(b) B”
- Line 366 & Line 393: In the Materials and Methods section, is it true that antibiotic doses are used in the concentration of 50 g/ml? This is a very high concentration. And also, rifampicin and kanamycin as 100 g ml-1 in Bacterial Matting section?
- Line 425-427 & 439-441: Reference No.1 is the same as No. 7.
- Line 69: the citation is missing, or change to “discovered in this study that …”
- What is the pE-CRP-1 construct? It is not mentioned in the Materials and Methods.
Author Response
May 18, 2023
The Reviewer,
International Journal of Molecular Sciences
Madam/Sir,
In the article " Unleashing the influence of cAMP Receptor Protein: The master switch of bacteriocin export in Pectobacterium carotovorum subsp. caratovorum" I'd like to express my gratitude for taking the time to review the aforementioned paper. Indeed, your suggestions are admirable, giving us optimism that our manuscript may be accepted in this in this prestigious publication.
In view of this, the following is a point-by-point response:
- Line 34: The citation in the first paragraph should be [1] or both [1] and [2].
- The arrangement of citation was corrected.
- Experimental procedures are not provided or clearly described, for example, how was the gene knockout performed? There is not too much information about the DNA plasmid construction of crp mutant (primers, restriction enzymes, etc.).
- The experimental procedures were re-written for clarity.
- Line 418: As this study involved the use of animals, should it include any institutional review board statement?
- Institutional review board statement was included with the date of approval written.
- Line 87-101: Figure 1 is of low quality, too blurry. There is no real gene sequence alignment except for just the statement of the alignment results, which can be removed from the figure and put into the paragraph. Just keep the alignment of amino acid sequences.
- The recommendation was followed, making the quality better.
- Line 116-130: In Figure 2, all number labels are too small, looks very blurry. And also, what are RH1-9? They should be clearly indicated for better understanding.
- The recommendation was followed.
- Line 170-174: In Figure 4 legend, the lane 2 in (a) is the same as the lane 1 in (b), but the SDS-PAGE gels show different bands. Please clearly describe/explain the difference. What is the big band in the lane 2 in (a)? And also, delete “B” in “(b) B”
- Figure 4, the observed band in lane 2 corresponds to pET32a, which produces a His-tag only. The presence of a prominent band can be attributed to overexpression after IPTG induction. In lane 3, the band represents the CRP-His fusion protein.
- Line 366 & Line 393: In the Materials and Methods section, is it true that antibiotic doses are used in the concentration of 50 g/ml? This is a very high concentration. And also, rifampicin and kanamycin as 100 g ml-1 in Bacterial Matting section?
- This was a typographical error. The section is corrected and micro was added.
- Line 425-427 & 439-441: Reference No.1 is the same as No. 7.
- Reference 7 was deleted.
- Line 69: the citation is missing, or change to “discovered in this study that …”
- The citation was change to "discovered in this….” as recommended.
- What is the pE-CRP-1 construct? It is not mentioned in the Materials and Methods.
pE-CRP-1 was already discussed in the added section in the materials and method about CRP purification and characterization.
Please find attached corrected manuscript for your reference.
Thank you and more power!
Yours Truly,
Duen-Yau Chuang, PhD
Corresponding Author
Round 2
Reviewer 1 Report
unfortunately, my suggestions (two) to revise the manuscript is pending.
If the author fails to generate mAb for CRP, I want the pAb characterization, and homology modeling, and show how the binding of pAb to the CRP. thereby we will know the epitope sequence for CRP protein.
same as above
Author Response
The Reviewer,
International Journal of Molecular Sciences
Madam/Sir,
Thank you for your valuable feedback on our research article titled " Unleashing the influence of cAMP Receptor Protein: The master switch of bacteriocin export in Pectobacterium carotovorum subsp. caratovorum". We sincerely appreciate your time and effort in reviewing our work and providing insightful comments.
Regarding your comment regarding the generation of monoclonal antibodies (mAbs) for CRP, we regret to inform you that we were unable to successfully generate mAbs in this particular study. However, we did perform a comprehensive characterization of the polyclonal antibody (pAb) targeting CRP. We have indeed performed comprehensive experiments to characterize the pAb and confirm its specificity for the CRP fusion protein. Additionally, we provided detailed information on the origin of the CRP protein, stating that it was derived from the full ORF region of CRP and subcloned into the pET32A vector and pE-CRP-1 overexpression vector. We also performed an analysis of the CRP and promoter binding region on Pcc's carocin, which we confirmed through a biotinylated probe precipitation experiment.
While we acknowledge the significance of homology modeling and epitope sequencing to understand the binding mechanism and epitope sequence of CRP, we would like to clarify that these analyses were beyond the scope of our current research objectives. Our focus was primarily on other specific aspects, as outlined in the article.
Nonetheless, we appreciate your suggestion, and we agree that homology modeling and epitope sequencing would be valuable for future investigations. We will duly consider incorporating these aspects in our future studies to provide a more comprehensive understanding of the CRP protein and its interactions.
Once again, we sincerely thank you for your time, expertise, and constructive feedback. Your comments have undoubtedly contributed to the improvement of our research article, and we are grateful for your support.
Thank you and more power!
Yours Truly,
Duen-Yau Chuang, PhD
Corresponding Author
Reviewer 2 Report
Dear Authors,
After careful revision of this version of the manuscript, I believe it is ready for publication.
Yours sincerely
Author Response
Dear Reviewer,
I am writing to express my deepest gratitude for your invaluable contribution as a reviewer for our research article. Your insightful feedback and constructive suggestions have played a crucial role in improving the quality and clarity of our work.
Your expertise and attention to detail have truly enhanced the overall integrity of the manuscript. We sincerely appreciate the time and effort you dedicated to thoroughly reviewing our research. Your thoughtful comments have not only helped us address the weaknesses in our study but also provided valuable perspectives that have significantly strengthened the paper.
Once again, we extend my heartfelt thanks for your dedication to maintaining the integrity of the scientific community through the peer-review process. Your expertise and commitment have been invaluable, and we are sincerely grateful for your time and effort.
Yours Truly,
Duen-Yau Chuang, PhD
Corresponding Author
Round 3
Reviewer 1 Report
I am sorry to say that the abstract of the work reads "The CRP binding sites were discovered using a biotinylated probe pull-down experiment and the study revealed that the deletion of crp inhibited genes involved in extracellular bacteriocin export" which means, the binding of CRP to its receptor should be determined or shown using computers.
I am sorry to say that the abstract of the work reads "The CRP binding sites were discovered using a biotinylated probe pull-down experiment and the study revealed that the deletion of crp inhibited genes involved in extracellular bacteriocin export" which means, the binding of CRP to its receptor should be determined or shown using computers.
Author Response
May 30, 2023
The Reviewer,
International Journal of Molecular Sciences
Madam/Sir,
Once again, we sincerely appreciate your thoughtful comments and valuable feedback on our research article titled " Unleashing the influence of cAMP Receptor Protein: The master switch of bacteriocin export in Pectobacterium carotovorum subsp. caratovorum".
You raised a concern regarding the abstract, specifically the statement that "The CRP binding sites were discovered using a biotinylated probe pull-down experiment, and the study revealed that the deletion of crp inhibited genes involved in extracellular bacteriocin export." You suggested that the binding of CRP to its receptor should be determined or demonstrated using computer-based methods.
We would like to inform that we have included additional statement in our abstract supported by the analysis presented in the Results section of our article. We mentioned in the abstract that "Analysis of the DNA sequence upstream of the translation initiation site of carocin S3 revealed two putative binding sites for CRP, which were confirmed using a biotinylated probe pull-down experiment." This statement indicates that we employed computer software to analyze the DNA sequence and identify the putative binding sites for CRP. Subsequently, we conducted a biotinylated probe pull-down experiment to confirm these predicted binding sites.
Therefore, the abstract accurately represents the sequence of our analysis, which involved a combination of computer-based analysis and experimental validation using the biotinylated probe pull-down technique.
We genuinely appreciate your attention to detail and your valuable comments, as they have allowed us to clarify this important aspect of our study. Your feedback has contributed significantly to the refinement of our research article, and we sincerely thank you for your time and expertise.
Yours Truly,
Duen-Yau Chuang, PhD
Corresponding Author